# *Aspergillus* Section *Fumigati* Pneumonia and Oxalate Nephrosis in a Foal

**DOI:** 10.3390/pathogens10091087

**Published:** 2021-08-26

**Authors:** Jasmine Hattab, Antonella Vulcano, Silvia D’Arezzo, Fabiana Verni, Pietro Giorgio Tiscar, Giovanni Lanteri, Emil Gjurcevic, Umberto Tosi, Giuseppe Marruchella

**Affiliations:** 1Faculty of Veterinary Medicine, University of Teramo, Loc. Piano d’Accio, 64100 Teramo, Italy; jhattab@unite.it (J.H.); pgtiscar@unite.it (P.G.T.); utosi@unite.it (U.T.); 2Laboratorio di Microbiologia Banca Biologica e Cell Factory, Istituto Nazionale Malattie Infettive “Lazzaro Spallanzani”, via Portuense 292, 00149 Rome, Italy; antonella.vulcano@inmi.it (A.V.); silvia.darezzo@inmi.it (S.D.); 3Veterinary Practitioner, Loc. Convento, 64023 Teramo, Italy; f.verni@yahoo.it; 4Department of Chemical, Biological, Pharmaceutical and Environmental Sciences, Viale G. Palatucci, 98168 Messina, Italy; giovanni.lanteri@unime.it; 5Department for Biology and Pathology of Fish and Bees, Faculty of Veterinary Medicine, University of Zagreb, Heinzelova 55, 10 000 Zagreb, Croatia; egjurcevic@vef.hr

**Keywords:** horse, pulmonary aspergillosis, *Aspergillus* section *Fumigati*, oxalate nephrosis

## Abstract

Equine pulmonary aspergillosis is a rare deep mycosis often due to the hematogenous spread of hyphae after gastrointestinal tract disease. We describe herein the main clinic-pathological findings observed in a foal, which spontaneously died after showing diarrhea and respiratory distress. Necropsy and histopathological investigations allowed to diagnose pulmonary aspergillosis, which likely developed after necrotic typhlitis-colitis. Biomolecular studies identified *Aspergillus* section *Fumigati* strain as the causative agent. Notably, severe oxalate nephrosis was concurrently observed. Occasionally, oxalate nephropathy can be a sequela of pulmonary aspergillosis in humans. The present case report suggests that the renal precipitation of oxalates can occur also in horses affected by pulmonary aspergillosis and could likely contribute to the fatal outcome of the disease.

## 1. Introduction

*Aspergillus* species are worldwide-distributed filamentous fungi (mold) which thrive in the soil as saprophytes and occasionally infect animals as opportunistic pathogens [1]. Equine aspergillosis is an uncommon but severe respiratory disease which can affect the sinonasal cavities, the guttural pouches or the lung parenchyma [2]. As a general rule, an immunosuppressive state and host-debilitating conditions predispose to fungal diseases, including aspergillosis. Horses usually become infected through the inhalation of fungal spores. Nevertheless, equine pulmonary aspergillosis (EPA) is often characterized by multifocal-to-disseminated embolic lesions which result from the hematogenous spread of hyphae after acute enteritis [2,3].

*Aspergillus* sp. can synthesize oxalic acid, which reacts with blood and tissue calcium to precipitate as oxalate crystals, at physiologic pH values [4]. In humans, the demonstration of such crystals in cytology specimens (e.g., in sputum) is useful to diagnosing *Aspergillus* infections, most commonly the *Aspergillus niger* group [5]. Likewise, the precipitation of oxalates within the foci of an *Aspergillus* infection has been reported in veterinary medicine (mostly in avian species). In a recent retrospective study, Payne et al. [6] observed the deposition of oxalates in 14 out of 38 cases of aspergillosis (*Aspergillus* sp., *Aspergillus fumigatus*, *Aspergillus niger*, *Aspergillus versicolor*), including 2 out of 15 horses.

We describe herein the peculiar clinic-pathological findings observed in a foal affected by pulmonary aspergillosis and oxalate nephrosis.

## 2. Case Description

The present case occurred in a Quarter Horse breed, 10-month-old male foal which belonged to a stable housing a total of 10 animals (five mares and five young horses less than two-years-old) in central Italy. The animal under study was kept in a single box provided with an external paddock and was fed with hay ad libitum and flaked feed. The foal was treated with a broad-range anthelmintic drug (ivermectin, Eqvalan^®^, Boehringer Ingelheim Animal Health, Milan, Italy) and vaccinated twice for influenza and tetanus at six and seven months of age (Equilis Prequenza Te^®^, MSD Animal Health, Milton Keynes, UK).

During the first decade of January 2020, the foal suddenly got prostrated, out of food and tachypneic. Within 24 h, the clinical picture severely worsened, with fever (42 °C), dyspnea, tachycardia and profuse watery diarrhea. At that time, the foal was treated with isotonic fluids (Ringer’s lactate, 15–20 L/die i.v.), antimicrobials (benzylpenicillin plus dihydrostreptomycin, Combiotic^®^, ACME srl, Cavriago, Italy, 10 mL/die i.m.), vitamin B complex (Dobetin B1-10000^®^, Ecuphar Italia srl, Milan, Italy 10 mL/die i.m.) and diosmectite (Smigol, ACME srl, Cavriago, Italy, 150 g/BID per os). However, the clinical signs further worsened, and the foal spontaneously died 48 h after the onset of the disease and was then necropsied.

A wide range of tissue samples (stomach, intestine, spleen, liver, kidneys, lungs, heart, lymph nodes) was promptly fixed in 10% neutral buffered formalin, embedded in paraffin and routinely processed for histopathological investigations (hematoxylin and eosin stain, H&E). Due to the precarious field conditions, no further sampling for microbiological investigations was carried out.

Grossly, the most relevant lesions affected the large intestine and the lungs. In more detail, the mucosa of the cecum and colon appeared necrotic, with abundant watery content filling their lumina. Microscopically, the intestinal mucosa was diffusely necrotic, infiltrated by inflammatory cells (mainly polymorphonuclear cells) and covered by bacterial aggregates; the lamina propria was strongly hyperemic, with scattered inflammatory cells seen also in this layer.

Both lungs appeared severely hyperemic and edematous. Moreover, disseminated nodules (2–5 mm in diameter) were appreciated at palpation throughout the lung parenchyma. Microscopically, pulmonary nodules appeared intensely hyperemic, the alveolar walls were thickened and congested. Foci of necrosis and hemorrhages were also seen, infiltrated by macrophages and polymorphonuclear cells. A huge number of branching septate fungal filaments, morphologically resembling *Aspergillus* sp., were detected inside the alveolar lumina, within the necrotic-hemorrhagic areas and the blood vessels (Figure 1). When subpleural nodules were investigated, the inflammatory reaction also involved the pleural surface, with numerous hyphae embedded within the fibrinous exudate which covered the visceral pleura.

Considering the microscopic findings, a formalin-fixed nodule was submitted for biomolecular study, aiming to identify the causative agent. A polymerase chain reaction assay targeting a highly conserved region of the 28S large ribosomal subunit (28S rDNA) was used as screening for the detection of clinically relevant fungi [7]. Section-level identification was achieved by sequence analysis followed by nucleotide BLAST analysis (https://blast.ncbi.nlm.nih.gov/Blast.cgi; accessed on: 29 July 2021). The results showed a 99% identity with *Aspergillus fumigatus* (GenBank: MH869824.1).

Remarkably, severe and extensive nephrosis was observed. Translucent and fan-shaped crystals filled almost all renal tubules, with no evidence of nephritis or fibrosis (Figure 2). The morphology of such crystals closely resembled that of calcium oxalates. The observation of H&E-stained sections under polarized light showed that crystals were strongly birefringent (Figure 3), thus confirming that they consisted of calcium oxalates. Careful examinations of the slides ruled out the deposition of oxalates in other tissues and organs.

Altogether, clinic-pathological and laboratory findings allowed us to diagnose *Aspergillus* section *Fumigati* strain pneumonia with concurrent severe oxalate nephrosis.

## 3. Discussion

EPA is widely recognized as a rare disease condition. In the largest case series currently available, Sweeney and Habecker [8] reported an estimated prevalence lower than 0.04%, i.e., 29 out of 73,000 horses admitted to a veterinary teaching hospital. Up to date, a total of 71 EPA cases have been recorded on the MEDLINE database (http://www.ncbi.nlm.nih.gov/pubmed/; accessed on: 16 July 2021) entering “horse, lung, aspergillosis” as keywords (see Table 1 for further details).

According to what was repeatedly reported in the literature, the present case of EPA most likely occurred after acute typhlitis-colitis, which disrupted the mucosal barrier of the large intestine and allowed the hematogenous spreading of fungi. The presence of disseminated lesions scattered throughout the parenchyma of both lungs, along with the detection of hyphae in the blood vessels, further support such a pathogenetic hypothesis. Slocombe and Slauson [3] hypothesized a synergistic relationship between acute enterocolitis by *Salmonella* sp. and EPA, although they often isolated different bacterial species from the intestine of horses affected by pulmonary aspergillosis (namely, *Escherichia coli*, *Klebsiella* sp., *Staphylococcus aureus*, *Clostridium perfringens*, *Proteus* sp., *Actinobacillus equuli*). In the present case report, pathological findings argue in favor of a bacterial etiology of typhlitis-colitis, which unfortunately could not be further investigated.

In veterinary medicine, oxalate nephrosis is frequently observed in dogs and cats due to ethylene glycol toxicity, as well as in small ruminants after the ingestion of plants of various genera (e.g., *Oxalis*, *Rumex*) which can accumulate oxalates. The production of oxalic acid is a common feature of filamentous fungi and is catalyzed by oxaloacetate hydrolase (OAH) [19]. In particular, some fungi (e.g., *Aspergillus niger*, *Aspergillus flavus*) can produce large amounts of oxalates on feedstuffs, which may represent another potential cause of poisoning [20]. *Aspergillus fumigatus* is less commonly recognized as an oxalate-producing pathogen [21]. It would be interesting to investigate the expression and activity of *Aspergillus* section *Fumigati* OAH to know whether differences exist in oxalate production among strains and/or under different culture conditions.

Generally, the horse is considered resistant to oxalate-induced nephrosis. In this animal species, acute poisoning results from exposure to very high dosage of oxalates in food and causes fatal gastroenteritis, while prolonged exposure to that chemical is known to induce osteodystrophia fibrosa [20]. Hypothetically, the food may have been the source of poisoning in the case described herein. However, this scenario is unlikely, considering that all the other horses living in the same stable and fed with the same hay and feedstuff did not show any clinical signs. Moreover, the subsequent analysis of the hay and feedstuffs demonstrated that the level of fungi and mycotoxin contamination was in the range of normality (data not shown).

Reasonably, we consider that oxalate nephrosis occurred as a secondary event after mycotic pneumonia. In this respect, the timing of pulmonary and renal lesions, which both developed in a short period, argues in favor of a direct pathogenetic link between an *Aspergillus* infection and oxalate deposition in renal tubules. Moreover, fatal cases of oxalate nephrosis occurred in humans after pulmonary *Aspergillus niger* group infection [22,23,24,25,26,27,28,29]. Likewise, cases of renal oxalosis have been rarely documented in wild ruminants, namely an elk (*Cervus elaphus nelsoni*) affected by *Aspergillus fumigatus* pneumonia [30] and a white tail deer (*Odocoileus virginianus*) with pneumonia and encephalitis caused by *Aspergillus fumigatus* [31].We remark that in wild ruminants, as well as in several human cases, the renal precipitation of oxalate crystals was not associated with the local infection by *Aspergillus* sp., and that oxalates were not always detected in lung lesions. Therefore, we speculate that oxalates, which are produced in the pulmonary foci of an *Aspergillus* infection, can reach the renal parenchyma through the bloodstream accumulating in the renal tubules, where the microenvironment is optimal for their precipitation.

As far as the horse is concerned, the presence of oxalate crystals has been reported only in two cases of aspergillosis (see Table 1 for details). In the first case, both fungi and oxalates were present in lungs and kidneys. In the second case, oxalates were seen in the renal parenchyma, while they remained undetected in the site of *Aspergillus* sp. infection (i.e., lung). In both cases, the renal deposition of oxalates was restricted to a single tubule, thus being of poor clinical significance [6].

## 4. Conclusions

The present case report supports that EPA should be suspected whenever a severe respiratory syndrome occurs after a gastrointestinal tract disease, particularly when antimicrobials prove to be ineffective [8]. *Aspergillus fumigatus* is the most commonly documented cause of EPA, although the identification of the causative agent often remains at the genus level. In this respect, biomolecular techniques provide useful tools in identifying the pathogen, even when a fungal culture cannot be performed. Finally, oxalate nephrosis can represent a severe, clinically relevant sequela of EPA—as occasionally observed in human medicine—thus likely contributing to the fatal outcome of the disease.

## Figures and Tables

**Figure 1 pathogens-10-01087-f001:**
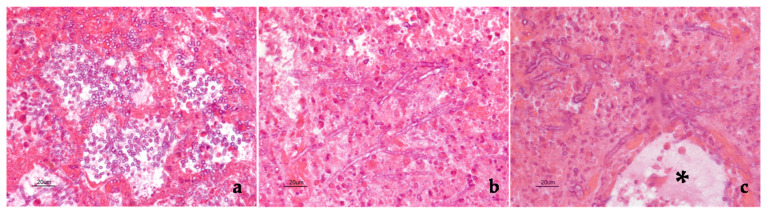
Foal. Pulmonary nodule. Several fungi were present inside the alveolar lumina (**a**), within necrotic-hemorrhagic foci (**b**) as well as in the blood vessel wall (black asterisk indicates the lumen of the same vessel). In some microscopic fields (**b**,**c**), dichotomous branching hyphae were seen. Hematoxylin and eosin stain. Final magnification 400×.

**Figure 2 pathogens-10-01087-f002:**
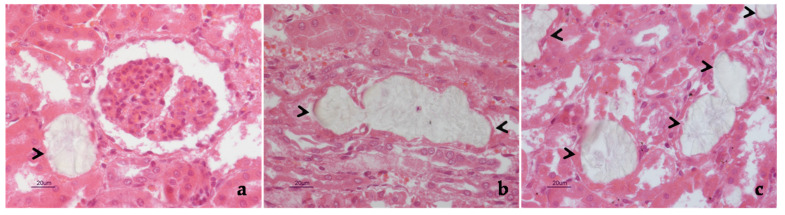
Foal. Kidney. Typical fan-shaped, oxalate crystals were seen in the lumina of renal tubules, both in transverse (**a**) and longitudinal (**b**) section (black arrowheads). The precipitation of oxalates affected most of renal tubules (**c**, black arrowheads), which appeared distended, their epithelium being no longer detectable (**a**–**c**). Hematoxylin and eosin stain. Final magnification 400×.

**Figure 3 pathogens-10-01087-f003:**
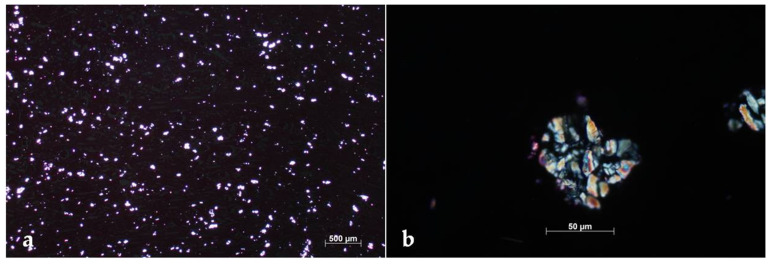
Foal. Kidney. At very low magnification, the precipitation of countless oxalate crystals was evident (**a**). At higher magnification, the typical birefringence, with radiating spokes, could be appreciated (**b**). Polarized light microscopy, hematoxylin and eosin stain. Final magnification 25× (**a**), 400× (**b**).

**Table 1 pathogens-10-01087-t001:** Cases of pulmonary aspergillosis so far reported in the horse.

Number of EPA Cases	Predisposing Factors	*Aspergillus* Species	Sites of Infection (Other Than Lung)	Sites of Oxalates Precipitation	Reference
19	Enterocolitis (*n* = 14), intensive corticosteroid treatment (*n* = 2), urolithiasis (*n* = 1), myelogenous leukemia (*n* = 1), hemangiosarcoma (*n* = 1)	*Aspergillus* sp. (*n* = 17), *Aspergillus fumigatus* (*n* = 2)	Kidneys (*n* = 2), brain (*n* = 1)	Not reported	[3]
11	Not reported	*Aspergillus* sp. (*n* = 11), *Aspergillus fumigatus* (*n* = 2), *Aspergillus glaucus* (*n* = 1), *Aspergillus flavus* (*n* = 1)	Colon (*n* = 1), kidney (*n* = 1), heart (*n* = 1), skeletal muscle (*n* = 1), guttural pouches (*n* = 1)	Lungs (*n* = 1) and kidneys (*n* = 2)	[6]
29	Loss of integrity of the gastrointestinal tract (*n* = 25)	*Aspergillus* sp.	Not specified organs (*n* = 12)	Not reported	[8]
2	None	*Aspergillus fumigatus*	Kidneys, myocardium, brain	Not reported	[9]
1	Cushing’s syndrome	*Aspergillus* sp.	None	Not reported	[10]
1	Leukemia	*Aspergillus* sp.	Gut	Not reported	[11]
1	Exposure to high spore count	*Aspergillus niger*	None	Not reported	[12]
1	*Ehrlichia risticii* infection, acute enteritis		None	Not reported	[13]
1	Volvulus of the colon and peritonitis	*Aspergillus* sp.	Heart	Not reported	[14]
2	*Sarcocystis neurona* infection (*n* = 1), small intestine obstruction (*n* = 1)	*Aspergillus* sp.	None	Not reported	[15]
1	Unknown	*Aspergillus fumigatus*	None	Not reported	[16]
1	Enterocolitis	*Aspergillus fumigatus*	Colon	Not reported	[17]
1	Myelomonocytic leukemia	*Aspergillus* sp.	None	Not reported	[18]

Gastrointestinal disorders appear as the most common predisposing factor (43 cases, i.e., 60.56%). The identification of the causative agent often does not go beyond the genus level; however, *Aspergillus fumigatus* is the most commonly identified species in EPA (8 cases, i.e., 11.26%). Oxalate crystals have been only reported in two horses.

## Data Availability

No new data were created or analyzed in this study. Data sharing is not applicable to this article.

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
