# Peer review of "Aspergillus Section Fumigati Pneumonia and Oxalate Nephrosis in a Foal"

_pathogens, 2021, doi:10.3390/pathogens10091087_

Round 1
Reviewer 1 Report
the authors describe a case o lung aspergillosis, colitis and oxalate nephrotoxicosis in a mare, giving their own interpretation of the possible pathogenesis. The main limit of this case report is that the pathogenesis can not be proven by the authors, thus remaining an hypotesis.
Authors proposed that A. fumigatus had its entry by the damaged intestinal mucosa and spread throught the vascular system. In this case the presence of fungal hyphae in different organs would have been expected.
Why authors did not performed a histochemical staining (Grocott, PAS) to better detect hyphae in tissues?
the primary cause of the colitis was not investigated. Why no microbiological analysis were performed? Do the authors think that Aspergillus was the primary cause of the intestinal flogosis?
these points should be further discussed in the text
Author Response
We really appreciate and understand the Rewiewer's comments.
COMMENT (1) The authors describe a case of lung aspergillosis, colitis and oxalate nephrotoxicosis in a mare, giving their own interpretation of the possible pathogenesis. The main limit of this case report is that the pathogenesis can not be proven by the authors, thus remaining an hypotesis.
ANSWER (1) We agree with the Reviewer. As detailed within the manuscript, the pathogenesis remains as a (plausible) hypothesis. However, we consider that is strongly supported by our findings and the literature, thanks to the previous publication of (few) similar cases in other species, including human beings.
COMMENT (2) Authors proposed that A. fumigatus had its entry by the damaged intestinal mucosa and spread throught the vascular system. In this case the presence of fungal hyphae in different organs would have been expected.
ANSWER (2) Actually, the entry of A. fumigatus by the intestinal mucosa is not our hypothesis. This scenario has been repeatedly described/suggested for equine pulmonary aspergillosis, as summarized in table 1. As correctly stated by the Reviewer, the presence of hyphae in different organs would be expected, after spreading through the bloodstream. However, the demonstration of hyphae only in the lungs is quite "common" EPA, their presence being occasionally reported in other tissue/organs (see Table 1 for details).
Comment (3) Why authors did not performed a histochemical staining (Grocott, PAS) to better detect hyphae in tissues?
ANSWER (3) As correctly stated by the Reviewer, histochemical staining are useful to better detect hyphae. That is especially true when few and/or faintly stained hyphae are seen within lesions in H&E slides (such conditions being quite common also in our experience). However, this is not the case. As shown in Figure 1 (lung parenchyma), a huge amount of hyphae were very easily detected, their morphological details (e.g. branching pattern) being also clearly appreciated in H&E slides. As far as different tissues are concerned, we remark that several slides from different paraffin blocks were observed; we neither detected any lesion compatible with fungal infections (hemorrhages, necrosis, inflammatory reactions), nor any structure resembling hyphae. Overall, in this case we considered of less value the performance of histochemical staining. Notwithstanding this, we are fully available to perform and add such tests, if necessary.
COMMENT (4) The primary cause of the colitis was not investigated. Why no microbiological analysis were performed? Do the authors think that Aspergillus was the primary cause of the intestinal flogosis?
ANSWER (4) We fully understand the Reviewer's comment. Unfortunately, as stated within the manuscript, necropsy was carried out under "poor" field conditions (and during "vacation time", at the beginning of January). That prevented a better and wider sampling. In addition, the horse under study had been treated with antimicrobials, which would have likely affected the results of bacterial culture tests. Last but not least, to be honest, I did not suspect such an interesting case at the time of necropsy (I had never experienced similar findings in horses). I/we do not think that Aspergillus caused intestinal flogosis. As reported in literature, cases of equine pulmonary aspergillosis often occur after acute bacterial enterocolitis, and this point is now further discusses within the manuscript.
Reviewer 2 Report
This is an interesting case of aspergillosis with concomitant oxalate nephrosis in a horse in Italy. Oxalate-producing Aspergillus infections do exist in both veterinary and medical literature but the number of cases remains very low and usually crystals are detected at the infection site. The present report is original because oxalate crystals secondary developed in the kidneys (without local Aspergillus infection).
The case is fairly well described and discussed. I suggest to reduce the number of figures. I also suggest to suppress table 1 (because most of the information included is related to cases of equine pulmonary aspergillosis without oxalate production... so without a direct interest for the present case). And here are specific questions/comments:
- Introduction : the first chapter should be suppressed (as well as ref 1 and 2). The manuscript should start with the following sentence: "Aspergillus species are..."
- Case description. I perfectly understand that "precarious field conditions" occurred and limited the possibility of biological investigations. However, I regret that there is no picture about the macroscopic aspect of the lesions.
- Microscopic pictures. Figures 1a, 2a, 2b and 3a should be suppressed. Why a Grocott's methenamine silver stain was not used? The sensitivity of this staining method is better than that of HES and it could have been useful to exclude the presence of hyphae in kidneys.
- Bacterial enteritis. The authors mentioned that the intestinal mucosa was covered by bacteria. Why didn't they try to identify the bacteria (through a biomolecular study for example)?
- Table 1 is not necessary and detailed information about the 2 equine cases with oxalate precipitation should be retrieved from the review of Payne et al. 2017
- Discussion. Aspergillus fumigatus is not a "classical" oxalate-producing Aspergillus species. The authors should discuss about the possibility that metabolic differences exist among A. fumigatus strains. May these potential differences be investigated through molecular analyses?
- Conclusion. The authors indicate that "oxalate nephropathy should be better investigated and carefully considered in the clinical management of equine pulmonary aspergillosis". Considering the EPA has a very poor prognosis (most of the cases are postmortem detection), I am not sure that this comment is really relevant...
- Final and general comments: latin names should be systematically written in italics. And "fungal hyphae" is a pleonasm: the authors shoud write: "fungal filaments" or "hyphae"
Author Response
We strongly appreciated all the Reviewer's comment, which have been addressed as follows.
COMMENT (1) Introduction : the first chapter should be suppressed (as well as ref 1 and 2). The manuscript should start with the following sentence: "Aspergillus species are..."
ANSWER (1) The manuscript has been changed as suggested by the Reviewer. As a consequence, references have been renumbered.
COMMENT (2) Case description. I perfectly understand that "precarious field conditions" occurred and limited the possibility of biological investigations. However, I regret that there is no picture about the macroscopic aspect of the lesions.
ANSWER (2) We also regret the absence of pictures showing gross lesions. Actually, that was mostly due to the features of lesions. As stated within the manuscript, lung nodules were easily appreciated at palpation. On the contrary, they were not easily seen (their hemorrhagic appearance within strongly hyperemic parenchyma made difficult their detection). We attempted to take pictures also later, in the lab, before and after fixation in formalin (sometimes fixation improves the contrast of colors), but we finally considered such photos of very poor quality (difficult-to-impossible to identify lesions).
COMMENT (3) Microscopic pictures. Figures 1a, 2a, 2b and 3a should be suppressed. Why a Grocott's methenamine silver stain was not used? The sensitivity of this staining method is better than that of HES and it could have been useful to exclude the presence of hyphae in kidneys.
ANSWER (3) As suggested by the Reviewer, microscopic pictures could appear redundant. However, each figure 1 (a-to-c) adds some useful information (the number of fungi, their branching, the involvement of blood vessels). Probably, a single figure 2 could be provided, but showing three pictures clearly highlights the severity of nephrosis. Moreover, we consider those pictures of high quality, useful for readers experiencing similar lesions. Overall, if possible, we would like to maintain pictures in the present format.
As suggested by both Reviewers, histochemical staining (PAS, Grocott) are commonly used to better detect fungi, as it is also in our routine diagnostic experience. That is especially true when few and/or faintly stained hyphae are present within lesions. In the present case, hyphae were very easily detected in H&E stained slides, their morphological features (e.g. branching pattern) being also clearly evident. Therefore we considered such staining of lesser value.
The deposition of oxalates in absence of "local" Aspergillus infection has been previously reported in literature. Should renal deposition of oxalates be due to renal Aspergillus infection, a huge number of hyphae would be expected in the kidneys. We carefully analyzed several tissue sections from different paraffin-embedded samples, thus we can reliably rule out the presence of hyphae in tissues other than the lung parenchyma. In addition, we highlight that we have never observed any microscopic lesion compatible with fungal infection (necrosis, hemorrhages, inflammation), apart from the lung parenchyma. However, we are fully available to stain additional sections with the PAS method, if necessary.
COMMENT (4) Bacterial enteritis. The authors mentioned that the intestinal mucosa was covered by bacteria. Why didn't they try to identify the bacteria (through a biomolecular study for example)?
ANSWER (4) As suggested also by the Reviewer 1, investigations about the etiology of enteritis would have been desirable to better detail the present case. Unfortunately, it was impossible to perform bacterial culture tests at the time of necropsy. Biomolecular investigations for fungi were (quite) easy and of high diagnostic value, thanks to the microscopic findings, which allowed to focus the target. We consider the same approach (although possible) would have been much more challenging and difficult to analyze for intestinal bacteria (presence of a huge number of species, also under healthy conditions, modulated by the administration of antimicrobials, likely invading the necrotic mucosa as secondary agents, etc). As an example, Slocombe and Slauson (not all the time) isolated Salmonella sp., Escherichia coli, Klebsiella sp., Staphylococcus aureus, Clostridium perfringens, Proteus sp., Actinobacillus equuli from EPA-affected horses. This point in now discussed in the manuscript.
COMMENT (5) Table 1 is not necessary and detailed information about the 2 equine cases with oxalate precipitation should be retrieved from the review of Payne et al. 2017.
ANSWER (5) We understand the reason for the Reviewer's comment. However, we consider this case report represents a useful opportunity to review the literature, aiming to "quantify" predisposing factors, fungal species involved, tissues/organs affected, frequency of oxalate deposition. This well beyond the retrospective (very interesting) study by Payne et a. (2017), which accounts for 15% of documented cases of equine pulmonary aspergillosis. Therefore, we would like to maintain Table 1 in the manuscript.
COMMENT (6) Discussion. Aspergillus fumigatus is not a "classical" oxalate-producing Aspergillus species. The authors should discuss about the possibility that metabolic differences exist among A. fumigatus strains. May these potential differences be investigated through molecular analyses?
ANSWER (6) We really appreciate this comment. This point is now discussed in the manuscript.
COMMENT (7) Conclusion. The authors indicate that "oxalate nephropathy should be better investigated and carefully considered in the clinical management of equine pulmonary aspergillosis". Considering the EPA has a very poor prognosis (most of the cases are postmortem detection), I am not sure that this comment is really relevant...
ANSWER (7) We agree with the Reviewer. Accordingly, the text has been modified (last sentence of the Discussion paragraph and Conclusion).
COMMENT (8)
Final and general comments: latin names should be systematically written in italics. And "fungal hyphae" is a pleonasm: the authors should write: "fungal filaments" or "hyphae"
ANSWER (8) The text has been corrected as indicated by the Reviewer. We commonly find the definition "fungal hyphae" in literature and "mechanically" use such terms, maybe influenced by the Italian habit to say "ife fungine". However, we fully agree with the Reviewer: it is a pleonasm! We corrected the manuscript and will treasure this right comment.